# Prime editing for functional repair in patient-derived disease models

Imre F. Schene[1,2,3,9], Indi P. Joore[2,3,9], Rurika Oka [4,5], Michal Mokry [1], Anke H. M. van Vugt[1,3], Ruben van Boxtel [4,5], Hubert P. J. van der Doef[6], Luc J. W. van der Laan [7], Monique M. A. Verstegen[7], Peter M. van Hasselt[2], Edward E. S. Nieuwenhuis[1,8,10] & Sabine A. Fuchs [1,2,3,10 ✉]

Prime editing is a recent genome editing technology using fusion proteins of Cas9-nickase and reverse transcriptase, that holds promise to correct the vast majority of genetic defects. Here, we develop prime editing for primary adult stem cells grown in organoid culture models. First, we generate precise in-frame deletions in the gene encoding β-catenin (*CTNNB1*) that result in proliferation independent of Wnt-stimuli, mimicking a mechanism of the development of liver cancer. Moreover, prime editing functionally recovers disease-causing mutations in intestinal organoids from patients with DGAT1-deficiency and liver organoids from a patient with Wilson disease (*ATP7B*). Prime editing is as efficient in 3D grown organoids as in 2D grown cell lines and offers greater precision than Cas9-mediated homology directed repair (HDR). Base editing remains more reliable than prime editing but is restricted to a subgroup of pathogenic mutations. Whole-genome sequencing of four prime-edited clonal organoid lines reveals absence of genome-wide off-target effects underscoring therapeutic potential of this versatile and precise gene editing strategy.

[1] Division of Pediatric Gastroenterology, Wilhelmina Children's Hospital, University Medical Center Utrecht, Lundlaan 6, 3584 EA Utrecht, the Netherlands. [2] Department of Metabolic Diseases, Wilhelmina Children's Hospital, University Medical Center Utrecht, Lundlaan 6, 3584 EA Utrecht, the Netherlands. [3] Regenerative Medicine Center Utrecht, Uppsalalaan 8, 3584 CT Utrecht, the Netherlands. [4] Princess Maxima Center, 3584 CS Utrecht, the Netherlands. [5] Oncode Institute, Princess Maxima Center, 3584 CS Utrecht, the Netherlands. [6] Department of Pediatric Gastroenterology, University Medical Center Groningen, Hanzeplein 1, 9713 GZ Groningen, the Netherlands. [7] Department of Surgery, Erasmus MC–University Medical Center Rotterdam, Doctor Molewaterplein 40, 3015 GD Rotterdam, the Netherlands. [8] Department of Sciences, University College Roosevelt, Lange Noordstraat 1, 4331 CB Middelburg, the Netherlands. [9] These authors contributed equally: Imre F. Schene, Indi P. Joore. [10] These authors jointly supervised this work: Edward E.S. Nieuwenhuis, Sabine A. Fuchs. ✉email: s.fuchs@umcutrecht.nl

The development of gene-editing therapies to treat monogenic diseases has long been an essential goal of CRISPR/Cas9 research. Cas9-mediated homology-directed repair (HDR) can create all desired base substitutions, insertions and deletions (indels). However, HDR relies on introduction of double-stranded DNA breaks, is inefficient and error-prone[1,2]. Base editing, that uses Cas9-nickases fused to DNA-modifying enzymes, is more efficient and accurate than HDR, but can only correct four out of twelve nucleotide substitutions and no small insertions and deletions. Furthermore, base editing requires a suitable protospacer adjacent motif (PAM) and the absence of co-editable nucleotides[3].

Prime editing combines a nicking-Cas9–reverse transcriptase fusion protein (PE2) with a prime editing guide RNA (pegRNA) containing the desired edit. The pegRNA-spacer guides the Cas9-nickase to create a nick in the targeted DNA strand. The pegRNA-extension binds to this nicked strand and instructs the reverse transcriptase to synthesize an edited DNA flap. This edited flap is then integrated by DNA repair mechanisms, which can be enhanced by simultaneous nicking of the non-edited strand (Supplementary Fig. 1)[4].

Prime editing has been applied in human cell lines, plant cells, and mouse embryonic cells but not in human disease models[4–7]. Adult stem cell-derived organoids exhibit important functional properties of organs, allowing modeling of monogenic diseases[8].

In this work, we develop prime editing in primary patient-derived organoids to show functional correction of disease-causing mutations and generation of representative disease models. We find that prime editing in 3D organoids is as efficient as in 2D cell lines and does not result in detectable genome-wide off-target effects.

## Results

**Prime editing efficiently creates mutations in organoids**. We first optimized prime editing for organoid cells using deletions and single-nucleotide substitutions previously performed in HEK293T cells[4]. The non-edited strand was nicked by a second "nicking sgRNA" to enhance editing (PE3). Our optimized protocol consisted of co-transfection of prime edit plasmids with a GFP-reporter plasmid allowing selection and subsequent clonal expansion of transfected cells. Prime editing of intestinal and ductal liver organoids resulted in efficient deletion of five nucleotides in HEK3, with the majority of picked clones containing monoallelic or biallelic deletions (Fig. 1b and Supplementary Fig. 2)[4]. Furthermore, we were able to induce a transversion mutation located 26 nucleotides downstream of the nick in 20% of the clones (Fig. 1c)[4].

Next, we targeted the Wnt-pathway intermediate β-catenin (CTNNB1) in organoids. Activating carcinogenic CTNNB1 mutations are found in ±40% of hepatocellular carcinoma, resulting in Wnt-signaling independent of exogenous stimuli[9]. We designed PE3 plasmids, containing pegRNA-extensions with primer binding sites (PBSs) and RT-templates of various lengths, that all create in-frame deletions in the β-TrCP region required for CTNNB1 ubiquitination (Fig. 1d). As wildtype liver organoid expansion depends on Wnt-pathway activation, edited cells could be selected by withdrawing Wnt-agonist R-spondin 1 (Fig. 1e). Sequencing confirmed that all clones grown without R-spondin for 2 weeks contained heterozygous in-frame deletions in CTNNB1 (Fig. 1f and Supplementary Fig. 2c). We observed striking differences (up to a factor 50) in editing efficiencies of different pegRNA designs (Fig. 1g). Next, we generated the severe ABCB11[D482G] mutation, a frequent cause of bile salt export pump (BSEP) deficiency[10]. This nucleotide substitution was generated in 20% of the liver organoid clones when silent PAM mutations

were introduced (Supplementary Fig. 3). These results demonstrate the utility of prime editing in creating disease models and the importance of testing different pegRNA designs to induce the desired edit.

To examine the efficiency and byproduct formation of prime editing in primary stem cells, we performed high-throughput sequencing of two targeted amplicons (HEK3 and CTNNB1). The desired edit was installed with 30–50% efficiency, while unwanted byproducts at the pegRNA or nickase sgRNA target sites only occurred at a rate of 1–4% in liver- and intestine-derived organoid cells. These rates were similar in two-dimensional HEK293T and Caco-2 cell lines from the same experiment (Fig. 1h). Distinct byproducts were shared between organoid lines and 2D cell cultures, suggesting that the mechanism of byproduct formation is independent of culture type (Supplementary Fig. 4). Together, these results show successful prime editing of primary stem cells with similar efficiency and accuracy as in human cancer cell lines.

**Prime editing functionally corrects disease-causing mutations.** To investigate prime editing for functional correction of disease-causing mutations, we studied *diacylglycerol-acyltransferase 1* (*DGAT1*) in patient-derived intestinal organoids. *DGAT1* encodes an enzyme catalyzing the conversion of diacylglycerol and fatty acyl-CoA to triacylglycerol. When DGAT1 function is deficient, fatty acids (FAs) cannot be incorporated in lipid droplets and instead cause lipotoxicity and cell death (Fig. 2a). *DGAT1* mutations result in congenital diarrhea and protein-losing enteropathy upon lipid intake[11]. The common biallelic 3-bp deletion (c.629_631delCCT, p.S210del) in exon 7 of *DGAT1* leads to complete absence of the mature protein (Fig. 2b, e)[11]. We designed PE3 plasmids to promote the insertion of the missing three nucleotides. These plasmids were transfected into patient-derived organoid cells and organoids were grown from single transfected cells. PE3 plasmids did not reduce the outgrowth efficiency or proliferation capacity of organoid cells, relative to a GFP plasmid only control (Supplementary Fig. 5a). Sanger sequencing of clonal organoids revealed repair of the pathogenic deletion (Fig. 2b). To demonstrate DGAT1 function of prime-edited cells, we exposed organoids to FAs, which are harmless to healthy control organoids and toxic to DGAT1-deficient organoids (Fig. 2c, d and Supplementary Fig. 5b). All clones surviving functional selection were genetically repaired (Fig. 2c and Supplementary Fig. 6a) and showed normal DGAT1 protein expression (Fig. 2e and Supplementary Fig. 6c). Next, we compared prime editing with HDR in terms of efficiency and accuracy. The ratio of correct editing to unwanted indels was ±30-fold higher for prime editing than for Cas9-initiated HDR (Fig. 2f and Supplementary Fig. 6d). These findings show that prime editing, as opposed to base editing, can repair small deletions with considerably higher precision and efficiency than HDR.

To compare prime editing to base editing, we selected two severe pathogenic G → A mutations suitable for correction by adenine base editors (ABEs): the BSEP-deficiency mutation *ABCB11*[R1153H] and the alpha-1 antitrypsin deficiency ZZ-genotype (*SERPINA1*[E342K])[10,12,13]. Without pegRNA design optimization, prime editing was outperformed by base editing in efficiency (Fig. 2g and Supplementary Fig. 7), indicating that the added value of prime editing currently lies in correcting mutations that are uneditable by base editors.

Next, we set out to repair a 1-bp duplication (c.1288dup, p.S430fs) in *ATP7B*, causing Wilson disease. *ATP7B* encodes a copper-transporter (ATP7B), facilitating excretion of excess copper into the bile canaliculus (Fig. 2h). Pathological

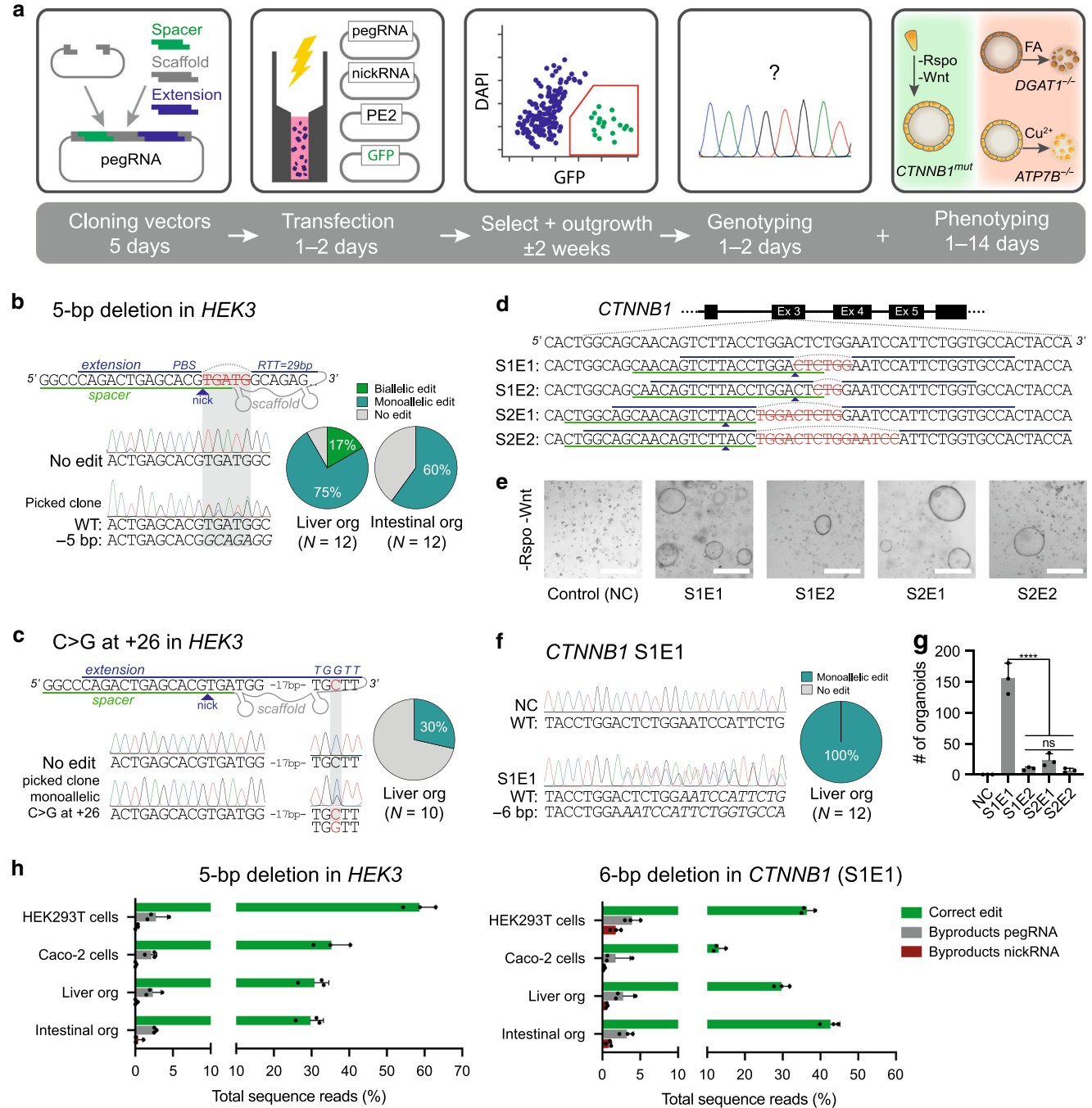

**Fig. 1 Prime editing efficiently creates deletions and point mutations in organoids. a** Schematic overview of the workflow and timeline to generate precise mutations in organoid cells using prime editing (PE3). **b** pegRNA design, Sanger validation in a clonal organoid with monoallelic edit, and editing efficiency of a 5-bp deletion in *HEK3* in liver and intestinal organoids. **c** pegRNA design, Sanger validation of monoallelic edit, and editing efficiency for a C → G substitution in *HEK3* in liver organoids. Nicking sgRNA at +90 used in **b** and **c** not shown. **d** pegRNA designs for the generation of in-frame deletions in the β-TrCP region of *CTNNB1*. Nicking sgRNA at +86 (S1) or +93 (S2) not shown. **e** Brightfield images of liver organoid cells transfected with plasmids from **d** after Rspo1 withdrawal for 2 weeks. White scale bars are 500 μm. **f** Sanger validation of precise 6-bp deletions in all picked clones from *CTNNB1* pegRNA S1E1 that continue growing in -Rspo1 conditions. **g** Quantification of organoid outgrowth in **e**. *p* < 0.0001 in a one-way ANOVA with Holm–Sidak correction. **h** Comparison of editing efficiencies and generation of unwanted byproducts in different cell types by high-throughput sequencing (HTS). Only transfected cells (GFP+ sorted) were used for HTS. Data are represented as mean values ±S.D. of three independent experiments **g** or biological replicates **h**. *FA* fatty acids, *PBS* primer binding site, *RTT* reverse transcriptase template, *NC* negative control. Source data are provided as a Source Data file.

accumulation of copper in the liver of Wilson disease patients leads to liver cirrhosis requiring lifelong treatment and ultimately liver transplantation[14]. We designed several PE3 conditions to remove the 1-bp duplication in patient-derived liver organoids (Supplementary Fig. 9). Clonal picking of transfected organoid

cells confirmed monoallelic repair of the disease-causing mutation by pegRNA#2 (Fig. 2i). We then generated *ATP7B*^KO organoids that were more susceptible to copper-induced cell death than *ATP7B*^WT organoids (Fig. 2j and Supplementary Fig. 7). To demonstrate functional repair of *ATP7B*^S430fs

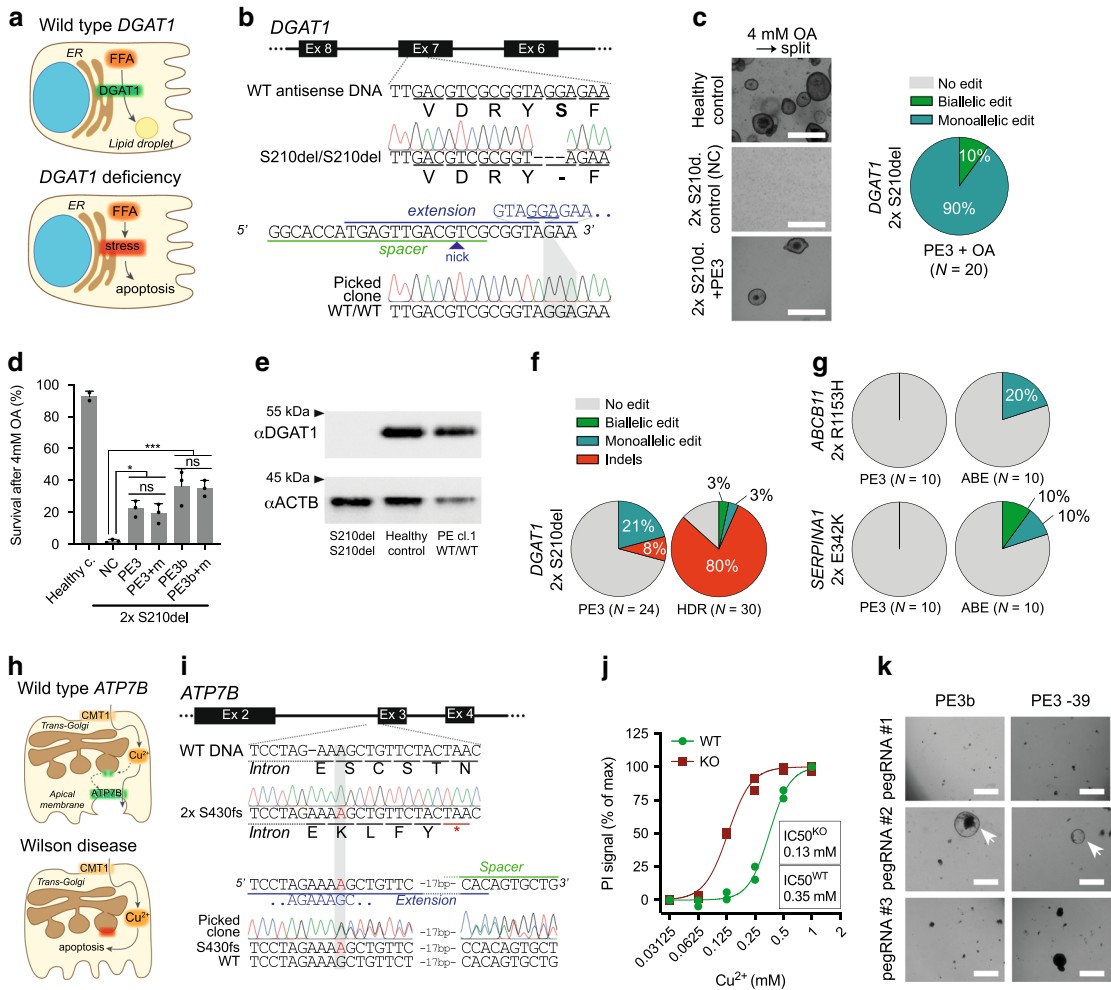

**Fig. 2 Prime editing functionally corrects disease-causing indel mutations in intestinal and liver organoids. a** Schematic overview of the DGAT1 disease mechanism. **b** Sanger validation of biallelic *DGAT1*^S210del mutations in patient-derived intestinal organoids, pegRNA design (nicking sgRNA at position +46 not shown), and Sanger validation of successful biallelic correction by PE3. **c** Brightfield images of healthy control- and *DGAT1*^S210del patient-derived intestinal organoids (±PE3) after exposure to 4 mM OA for 24 h and subsequent passaging (split). White scale bars are 500 μm. Quantification of corrected alleles in patient organoids after PE3 and OA selection; all surviving organoids are gene-corrected. **d** Quantification of *DGAT1*^S210del patient organoid survival upon exposure to 4 mM OA, after targeting with different PE3 or PE3b plasmids. Data are represented as mean ±S.D. of three independent experiments in two different donors. **e** Western blot of DGAT1 in *DGAT1*^S210del, healthy control, and PE3-corrected *DGAT1*^S210del organoids. **f**, Quantification of correct edits and unwanted indels by PE3 and Cas9-initiated HDR in *DGAT1*^S210del organoids. Note that no functional selection with OA was performed prior to quantification. **g** Comparison of PE3 and adenine base editing (ABEmax-NG) to correct the *ABCB11*^R1153H and *SERPINA1*^E342K mutations in liver organoids from patients with BSEP-deficiency and alpha-1-antitrypsin deficiency, respectively. **h** Schematic overview of the Wilson disease (ATP7B deficiency) mechanism. **i** Sanger validation of biallelic *ATP7B*^S430fs mutations in patient-derived liver organoids, pegRNA#2 design to target this mutation, and Sanger validation of successful monoallelic correction by PE3. **j** Cell death in *ATP7B*^WT and *ATP7B*^KO liver organoids after incubation with Cu²⁺ for 3 days. n = 2 biologically independent samples for both WT and KO groups. **k** Brightfield images of *ATP7B*^S430fs-patient organoid survival upon exposure to 0.25 mM Cu2+ for 3 days, after transfection with different PE3 plasmids. "at −39" stands for nicking sgRNA at position −39. Note that only prime editing using pegRNA#2 yields functional correction of *ATP7B* (white arrowheads). White scale bars are 500 μm. *ER* endoplasmic reticulum, *FFA* free fatty acid, *Ex* exon, *NC* negative control, *OA* oleic acid, *PE3+ m* PE3 with introduction of PAM mutation, *HDR* homology-directed repair, *ABE* adenine base editor, *PI* propidium iodide. Source data are provided as a Source Data file.

organoids after prime editing, transfected cells were exposed to copper for 4 days (Supplementary Fig. 2). Out of three different pegRNA plasmids, only successful monoallelic repair with pegRNA#2 resulted in rescue of copper excretion (Fig. 2k). These results confirm the ability of prime editing to genetically and functionally correct truncating mutations and underline the importance of testing various pegRNA designs.

**Prime editing induces no genome-wide off-target effects.** The rate at which genome editors generate undesired mutations across

the genome is a major determinant of their therapeutic potential. To date, no genome-wide examination of the fidelity of prime editors has been conducted in human cells. We therefore performed whole-genome sequencing (WGS) analysis on two prime-edited clones and their respective unedited control clones for both the *CTNNB1* 6-bp deletion in liver organoids and the *DGAT1* 3-bp insertion in intestinal organoids. To identify possible variants induced by prime editing, mutational profiles of clones were background-corrected for variants already present in the donor bulk culture (Fig. 3a). At in silico predicted off-target sites (204 and 287 for *CTNNB1* and *DGAT1* edits, respectively) no

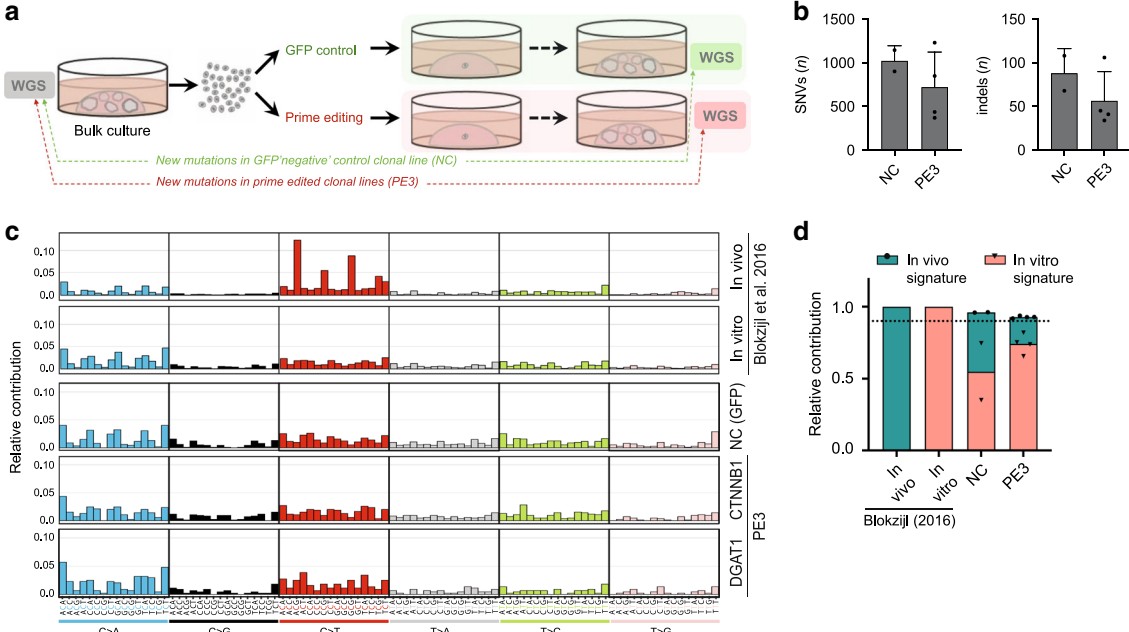

**Fig. 3 Prime editing induces no genome-wide off-target effects. a** Schematic overview of the protocol used to identify mutations induced by prime editing (PE3). WGS was performed for one unedited negative control and two prime-edited clonal lines for both DGAT1 (intestinal organoids) and CTNNB1 (liver organoids). **b** Total number of single-nucleotide variants (SNVs) and insertions and deletions (indels) in control (NC) and prime-edited (PE3) clonal organoid lines ($n = 2$ and $n = 4$ biologically independent samples, respectively). Error bars represent S.D. **c** Mutational signature analysis by relative contribution of context-dependent mutation types in an in vivo ($n = 6$) and in vitro ($n = 6$) data set (Blokzijl et al.[16]) and in control and prime-edited clonal organoid lines. **d** Relative contribution of known in vivo and in vitro mutational signatures from Blokzijl et al.[16] in control (NC) and prime-edited (PE3) clonal organoid lines ($n = 2$ and $n = 4$ biologically independent samples, respectively). Dotted line at 0.9 indicates highly accurate signature reconstruction. NC negative control, WGS whole-genome sequencing.

mutations occurred in a range of 200 bp in any of the prime-edited clones (Supplementary Table 4). The total number of new base substitutions or indels was not higher in prime-edited clones compared to controls (Fig. 3b). Specifically, no additional 6-bp deletions or 3-bp insertions appeared (Supplementary Table 4). Furthermore, no clustering of new mutations on a genome-wide scale, known as mutational hotspots, was present in any of the samples (Supplementary Fig. 10b).

Ongoing mutational accumulation during cell culture prevents direct attribution of specific mutations to prime editing. We therefore applied mutational signature analysis to search for mutational patterns due to potential aberrant prime editor activity[15,16]. The mutational signatures of prime-edited clones and unedited negative controls (NCs) were highly similar, both resembling the signature that arises during long-term propagation of intestinal organoids in vitro (Fig. 3c)[15]. Signatures of NCs and prime-edited samples could be reconstructed to a comparable degree by the combination of known in vivo- and in vitro mutational signatures (cosine similarity 0.92–0.96; Fig. 3d and Supplementary Fig. 10c). This indicates that prime editors do not leave a mutational fingerprint at the genome-wide scale. Safety of prime editing was further confirmed by absence of additional oncogenic mutations in tumor suppressor genes or oncogenes compared to negative controls, apart from the intended 6-bp deletion in CTNNB1 samples (Supplementary Fig. 10d)[17].

## Discussion

Versatile, efficient, and safe gene editing in primary cells represents a gamechanger for both in vitro modeling of monogenetic diseases and treatment with autologous gene-corrected cells. Here, we provide a protocol for effective prime editing in human adult stem cells. Using this protocol, we demonstrate that prime editing can generate insertions, deletions, and various point

mutations and functionally correct disease phenotypes in patient-derived stem cells. We find high editing rates and low byproduct formation in both 3D-cultured adult stem cell organoids and 2D-cultured cancer cell lines. Importantly, prime editing results in higher efficiency and drastically lower indel-formation compared with Cas9-mediated HDR. In the subset of mutations applicable to base editing, prime editors are currently less robust than the latest generation of base editors.

Our data in primary adult stem cells corroborate findings in cancer cell lines and mouse cortical neurons, in which prime editing typically offers greater precision than Cas9-mediated HDR and lower on-target efficiency than base editing, when the target nucleotide is suitably located[4]. Base editing, however, has been subject to various stages of optimization, whereas prime editing is still in its infancy[13,18,19]. The efficiency of prime editing in mammalian and plant cells has been strongly associated with pegRNA design, but optimal PBS and RT-template parameters remain elusive and differ between target sites[4,6,7]. In our hands, pegRNA design also had profound effects on editing efficiency, with PBS lengths of 10–12 nucleotides outperforming longer designs (Figs. 1g and 2k). We expect that pegRNA design as well as the prime editor fusion protein can be further optimized to improve prime editing efficiency in the future.

The WGS analysis provided here constitutes the most comprehensive genome-wide interrogation of prime editor fidelity to date. We did not find any off-target effects at locations resembling the target site. Neither could we identify a mutational signature induced by prime editor enzymes. Despite our small sample size, this absence of genome-wide off-target effects is reassuring for further therapeutic development.

To conclude, this study confirms the potential of prime editing to model and safely repair human monogenic diseases and represents an important step towards future clinical application.

## Methods

**Study approval and human subjects**. The study was approved by the responsible local ethics committees (Institutional Review Board of the University Medical Center Utrecht (STEM: 10-402/K) and Erasmus MC Medical Ethical Committee (MEC-2014-060). Tissue biopsies from liver of a patient with BSEP-deficiency was obtained during a liver transplant procedure in the UMCG, Groningen, and biopsies from duodenum of two patients with DGAT1-deficiency was obtained during diagnostic duodenoscopy in the UMCG, Groningen.

Tissue biopsies from livers of patients with Wilson disease and alpha-1 antitrypsin deficiency were obtained during liver transplant procedures in the Erasmus MC, Rotterdam. All biopsies were used after written informed consent.

**Organoid culture**. Liver and intestinal organoids were cultured and passaged according to previously described protocols[12,20]. In short, liver organoids were plated in matrigel (Corning) and maintained in liver expansion medium (EM), consisting of AdDMEM/F12 (GIBCO) supplemented with 2% B27 without vitamin A (GIBCO), 1.25 mM N-Acetylcysteine (Sigma), 10 mM Nicotinamide (Sigma), 10 nM gastrin (Sigma), 10% RSPO1 conditioned media (homemade), 50 ng/ml EGF (Peprotech), 100 ng/ml FGF10 (Peprotech), 25 ng/ml HGF (Peprotech), 5 mM A83-01 (Tocris), and 10 mM FSK (Tocris). SI organoids were plated in matrigel and maintained in SI EM, consisting of AdDMEM/F12 supplemented with 50% WNT3A-, 20% RSPO1-, and 10% NOG(gin)-conditioned medium (all home-made), 2% B27 with vitamin A (GIBCO), 1.25 mM N-Acetylcysteine, 10 mM Nicotinamide, 50 ng/ml murine-EGF (Peprotech), 500 nM A83-01, and 10 mM SB202190 (Sigma). The medium was changed every 2–4 days and organoids were passaged 1:4 –1:8 every week. After thawing, organoids were passaged at least once before electroporation.

**Plasmid cloning**. Cloning of pegRNA plasmids was performed according to previously described protocols[4]. In brief, the pU6-pegRNA-GG-Vector (Addgene #132777) was digested overnight with BsaI-HFv2 (NEB) and the 2.2 kb fragment was isolated. Oligonucleotide duplexes containing the desired pegRNA spacer, pegRNA extension, and pegRNA scaffold sequences were ordered with the appropriate overhangs and annealed. Annealed pegRNA duplexes were cloned into the pU6-pegRNA-GG-Vector using Golden Gate assembly with BsaI-HFv2 (NEB) and T4 DNA ligase (NEB) in a protocol of 12 cycles of 5 min at 16 °C and 5 min at 37 °C. For cloning of sgRNAs used for PE3 and ABE-NG, we replaced the BsmBI restriction sites of the sgRNA expression vector BPK1520 by BbsI restriction sites using PCR, which allowed direct ligation of sgRNA-spacer duplexes[21]. All pegRNA, sgRNA, HDR template, and primers sequences used in this work are listed in Supplementary Tables 1–3 and were synthesized by Integrated DNA Technologies. pCMV-PE2 (Addgene #132775), pU6-pegRNA-GG-acceptor (Addgene #132777), and NG-ABEmax (Addgene #124163) were gifts from David Liu; BPK1520 (Addgene #65777) was a gift from Keith Joung; pSpCas9(BB)-2A-Puro (PX459, Addgene #62988) was a gift from Feng Zhang.

**Electroporation**. Before electroporation, organoids were grown under normal conditions in 30 μl Matrigel per well. Two days prior to electroporation, organoids were cultured in medium containing surrogate WNT protein (4 nM). Four wells containing organoids were then dissociated for each condition using TrypLE for 4–5 min at 37 °C and applying mechanical disruption through pipetting. Cells were washed once using Advanced DMEM/F12, resuspended in 80 μl OptiMEM containing Y-27632 (10 μM), and 20 μl DNA mixture was added. For prime editing, the DNA mixture contained 12 μg PE2 plasmid, 5 μg pegRNA plasmid, 2 μg nicking sgRNA plasmid, and 1 μg GFP plasmid. For HDR, the DNA mixture contained 15 μg sgRNA containing-Cas9 plasmid (PX459), 1 μl of 100 μM HDR template, and 1 μg GFP plasmid. For base editing, the DNA mixture contained 15 μg NG-ABEmax plasmid, 4 μg sgRNA plasmid, and 1 μg GFP plasmid. For generation of ATP7B knockout organoids, the DNA mixture contained 15 μg sgRNA containing-Cas9 plasmid (PX459) and 1 μg GFP plasmid. The cell-DNA mixture was transferred to an electroporation cuvette and electroporated using a NEPA21 electroporator (NEPA GENE) with 2× poring pulse (voltage: 175 V, length: 5 ms, interval: 50 ms, polarity: +) and 5× transfer pulse (voltage: 20 V, length: 50 ms, interval: 50 ms, polarity ±), as previously described[22]. Cells were removed from the cuvette and transferred into 500 μl OptiMEM containing Y-27632 (10 μM). After 20 min, cells were plated in 180 μl matrigel. Upon polymerization of the Matrigel, medium was added containing surrogate WNT protein (4 nM) and Y-27632 (10 μM).

**Fluorescence-activated cell sorting (FACS)**. After 2–3 days of electroporation using the GFP plasmid, cells were dissociated with TrypLE for 2–3 min at 37 °C. The cells were washed once using Advanced DMEM/F12 and resuspended in 400 μl fluorescence-activated cell sorting (FACS) buffer (phosphate-buffered saline with 2 mM ethylenediaminetetraacetic acid and 0.5% bovine serum albumin). FACS was used to filter the GFP+ cell population, selecting specifically for transfected cells (FACS Aria, BD). GFP+ cells were retrieved in medium containing surrogate WNT protein (4 nM) and Y-27632 (10 μM), after which the cells were plated as soon as possible at a cell-concentration of ±300 cells per 30 μl Matrigel. Upon polymerization of the Matrigel, medium was added containing surrogate WNT protein (4 nM) and Y-27632 (10 μM).

**Organoid reconstitution and proliferation**. The number of organoids in each condition were counted by an automated counting algorithm 7 days after seeding single FACS-sorted cells. Organoid reconstitution was calculated as percentage of (number of organoids at day 7/number of cells seeded at day 0). Organoid cell proliferation was measured by quantification of average organoid size at day 7 after seeding single cells. At measurements, organoids were incubated with 1 μM Cell-Trace Calcein Green AM (Thermo Scientific) for 30 min and subsequently imaged by an inverted Olympus IX53 epifluorescence microscope (Tokyo, Japan). Images were analyzed using an automated organoid counting algorithm written in ImageJ and average organoid size was calculated for each condition and normalized to the control condition (GFP only).

**Genotyping**. Single organoids were picked using a p200 pipette and dissociated using TrypLE for 2–3 min at 37 °C. Cells were resuspended in 30 μl Matrigel total of which 20 μl was plated. DNA was extracted from the remaining 10 μl Matrigel using the Zymogen Quick-DNA microprep kit according to manufacturer instructions. Q5 high fidelity polymerase was used to amplify the genomic region of interest. The PCR product was purified using the Qiagen PCR clean-up kit according to manufacturer instructions. Resulting product was sent for Sanger sequencing to the Macrogen Europe service EZSeq.

**High-throughput DNA sequencing of genomic DNA samples**. Genomic sites of interest were amplified from genomic DNA samples and sequenced on an Illumina MiSeq as previously described[4]. In brief, Illumina forward and reverse adapters (Supplementary Table 3) were used for a first round of PCR (PCR1) to amplify the genomic region of interest. In a second round of PCR (PCR2) each sample was barcoded with unique Truseq DNA Index primers (Illumina). DNA concentration was measured by fluorometric quantification (Qubit, ThermoFisher Scientific) and sequenced on an Illumina MiSeq instrument according to the manufacturer's protocols. Sequencing reads were demultiplexed using MiSeq Reporter (Illumina). Alignment of amplicon sequences to reference sequences was performed by Cas-analyzer in HDR mode, using the unedited sequence as the reference sequence and the desired sequence as HDR donor DNA sequence[23]. Prime editing efficiency was calculated as the percentage of (number of reads with the desired edit/number of total aligned reads). For unwanted byproduct analysis at the pegRNA or nickase sgRNA site, a comparison range (R) of 30 bp was used so that 60 bp flanking the predicted nicking site were considered. Frequency of byproducts was calculated as the percentage of (number of reads with unwanted edits/number of total aligned reads).

**Protein blotting**. Organoids were lysed in Laemmli buffer (0.12 M Tris-HCl pH 6.8, 4% SDS, 20% glycerol, 0.05 g/l bromophenol blue, 35 mM β-mercaptoethanol). Protein concentration was measured using a BCA assay. For western blotting, equal amounts of protein were separated by SDS-PAGE on a 10% acrylamide gel and transferred to polyvinylidene difluoride (PVDF) membranes using a Trans-Blot® Turbo machine (Bio-rad) according to manufacturer's protocol. For dot blotting, protein was directly loaded on PVDF membranes without SDS-PAGE separation. The membrane was blocked with 5% milk protein in tris-buffered saline with Tween 20 (0.3% Tween, 10 mM Tris-HCl pH 8 and 150 mM NaCl in distilled water) and probed with primary antibodies against DGAT1 (ab181180; 1:1000; Abcam) or ACTB (sc-47778; 1:5000, Santa Cruz Biotechnology) overnight at 4 °C. After incubation with horseradish peroxidase (HRP)-conjugated secondary antibodies (1:5000, DAKO p0260 and p0217, 1 h at RT), bands or dots were imaged on a chemiluminescence detection system (Bio-rad).

**Functional assays**. To select liver organoids with in-frame mutations in *CTNNB1*, organoids were cultured in normal culture medium without R-spondin 1 for 2 weeks. To test DGAT1 function in intestinal organoids, 4 mM oleic acid was added to the culture medium for 24 h and organoid survival was assessed by visual inspection and survival after passaging. To test ATP7B function in liver organoids, copper(II)chloride (CuCl$_2$) was added to the culture medium for 3–4 days. Cell death was quantified by addition of propidium iodide (PI) (0.1 mg/mL, Thermo-Fisher) to the culture medium for 15 min. Organoids were imaged by an inverted Olympus IX53 epifluorescence microscope. PI signal was quantified using ImageJ and normalized to a positive control condition for cell death (1 mM CuCl$_2$). Based on *ATP7B*$^{KO}$ lines, prolonged organoid survival in 0.25 mM CuCl$_2$ was considered as a characteristic of functional ATP7B.

**WGS and mapping**. Genomic DNA was isolated from ±5 × 10$^5$ cells using the Zymogen Quick-DNA microprep kit according to manufacturer's instructions. Standard Illumina protocols were applied to generate DNA libraries for Illumina sequencing from 20 to 50 ng of genomic DNA. All samples (two genetically corrected clones, one non-corrected control sample, and one "bulk" samples from the starting culture for both the CTNNB1 6-bp-deletion and DGAT1 3-bp-insertion) were sequenced (2 × 150 bp) using Illumina NovaSeq to 30× base coverage. Reads were mapped against human reference genome hg19 using Burrows-Wheeler Aligner v0.5.9[24] with settings "bwa mem -c 100 -M". Duplicate sequence reads were marked using Sambamba v0.4.7.32 and realigned per donor using Genome

Analysis Toolkit (GATK) IndelRealigner v2.7.2 and quality scores were recalibrated using the GATK BaseRecalibrator v2.7.2. More details on the pipeline can be found on Github[25].

**Mutation calling and filtering**. Raw variants were multisample-called by using the GATK HaplotypeCaller v3.4–46[26] and GATK- Queue v3.4–46 with default settings and additional option "EMIT_ALL_CONFIDENT_SITES". The quality of variant and reference positions was evaluated by using GATK VariantFiltration v3.4–46 with options "-snpFilterName LowQualityDepth -snpFil- terExpression "QD < 2.0" -snpFilterName MappingQuality -snpFilterExpression "MQ < 40.0" -snpFilter-Name StrandBias - snpFilterExpression "FS > 60.0" -snpFilterName Haploty-peScoreHigh -snpFilterExpression "HaplotypeScore > 13.0" -snpFilter- Name MQRankSumLow -snpFilterExpression "MQRankSum < 12.5" -snpFilterName ReadPosRankSumLow -snpFilterExpression "ReadPosRankSum <8.0" -cluster 3 -window 35". To obtain high-quality somatic mutation catalogs, we applied post processing filters as described[16]. Briefly, we considered variants at autosomal chromosomes without any evidence from a paired control sample ("bulk" starting culture); passed by VariantFiltration with a GATK phred-scaled quality score R 250; a base coverage of at least 20× in the clonal and paired control sample; no overlap with single-nucleotide polymorphisms in the Single Nucleotide Poly-morphism Database v137.b3730; and absence of the variant in a panel of unmat-ched normal human genomes (BED-file available upon request). We additionally filtered base substitutions with a GATK genotype score (GQ) lower than 99 or 10 in the clonal or paired control sample, respectively. For indels, we filtered variants with a GQ score lower than 99 in both the clonal and paired control sample and filtered indels that were present within 100 bp of a called variant in the control sample. In addition, for both SNVs and INDELs, we only considered variants with a mapping quality score of 60 and with a variant allele frequency of 0.3 or higher in the clones to exclude in vitro accumulated mutations[16]. The scripts used are available on Github[27]. The distribution of variants was visualized using an in house developed R package (MutationalPatterns)[16].

**In silico off-target prediction**. Potential sgRNA specific off-target events were predicted using the Cas-OFFinder open recourse tool[28]. All potential off-targets up to four mismatches were taken into account.

**Mutational signature analysis**. We extracted mutational signatures and estimated their contribution to the overall mutational profile as described using an in house developed R package (MutationalPatterns)[16]. In this analysis, we included small intestine data (previously analyzed)[15] to explicitly extract in vivo and in vitro accumulated signatures[16].

**Statistics and reproducibility**. No pre-specified effect size was calculated, and no statistical method was used to predetermine sample size. The source data for figures can be found in the Source Data file. For comparisons of multiple groups, an ordinary one-way analysis of variance with Holm–Sidak correction for multiple comparisons was used and performed in Prism (GraphPad Software). All graphs were plotted using Prism (GraphPad Software). In Figs. 1g and 2d the negative control and the healthy control, respectively, were excluded from statistical com-parisons. Statistical tests were appropriate for comparisons being made; assessment of variation was carried out but not included. Experiments were not randomized. Investigators were not blinded to allocation during experiments but outcome assessment (sequencing and functional assay quantification) were performed blinded. Reproducibility: Fig. 1b representative of 12 and 10 clonal liver and intestinal organoids, respectively, each from two independent transfection experi-ments. Figure 1c, f representative of 10 clonal liver organoids, from two and three independent transfection experiments, respectively. Figure 1g representative of three transfection experiments using the same liver organoid donor. Figure 1h representative of three biological replicates from one transfection experiment. Figure 2c representative of 20 clonal organoids from two independent experiments. Figure 2d representative of three independent transfection experiments in two different donors. Figure 2e and Supplementary Fig 6c represent a single blotting experiment. Figure 2f representative of 24 and 32 clonal organoids from two independent experiments. Figure 2g representative of 10 clonal liver organoids per condition, collected through two independent experiments. Figure 2j represents a single experiment. Figure 2k representative of two experiments. Figure 3 and Supplementary Fig. 10 represent a single experiment with two edited clones and one negative control for each of two prime edits. Supplementary Fig. 3 repre-sentative of 30 clonal liver organoids from two independent experiments. Sup-plementary Fig. 5b are representative images for three independent transfection experiments.

**Reporting summary**. Further information on research design is available in the Nature Research Reporting Summary linked to this article.

## Code availability

Cas-Analyzer is publicly available [http://www.rgenome.net/cas-analyzer]. The algorithms used for mapping [https://github.com/UMCUGenetics/IAP], mutational calling [gatk.broadinstitute.org], mutational filtering [https://github.com/ToolsVanBox/SMuRF], and mutational pattern analysis [https://www.bioconductor.org/packages/release/bioc/html/MutationalPatterns.html] of WGS data are all publicly available. Cas-OFFinder was used for in silico prediction of off-target sites of pegRNA and sgRNA spacers and is available at [http://www.rgenome.net/cas-offinder/]. The algorithms to quantify the number and size of organoids, as well as to quantify the signal of propidium iodide, were written in ImageJ and are available from the corresponding author on reasonable request.

## Data availability

Source data for the figures have been provided as a Source Data file. The WGS samples of Fig. 3 and Supplementary Fig. 10 have been submitted to the European Genome-phenome Archive under study number EGAS00001004611. All other data and material supporting the findings of this study are available from the corresponding author on reasonable request. Source data are provided with this paper.

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

## Acknowledgements

The authors are grateful for the collaborative "United for Metabolic Diseases (UMD)" efforts to improve care for patients with (genetic) metabolic diseases. We thank M. Geurts (Hubrecht Institute, the Netherlands) for providing WNT surrogate and for sharing his experiences with prime editing. We thank Henkjan J. Verkade sharing unique patient-derived material. This work was supported by Metakids funding (to S.A.F.) and a Clinical Fellows grant from The Netherlands Organisation for Health Research and Development Health Institute (40-00703-97-13537 to S.A.F.).

## Author contributions

I.F.S., I.P.J., R.B., E.E.S.N., and S.A.F. designed the project; H.P.J.D., L.J.W.K., and M.M.A.V. helped establishing the biobank of patient-derived stem cell organoids used in this study, I.F.S., I.P.J., and A.J.M.v.V. performed experiments and analysis; R.O. performed analyses; I.F.S., I.P.J., M.M., P.M.v.H., E.E.S.N., and S.A.F. wrote the manuscript.

## Competing interests

The authors declare no competing interests.
