## [Peer Review File · Nature Communications]

Reviewers' Comments:

Reviewer #1:

Remarks to the Author:

The manuscript by Schene and Joore describes adaptation of the "prime editing" method for human adult stem cell derived organoids. This provides an exciting and versatile new possibility for gene correction and gene targeting in primary human epithelia. The methodology has only reported last year and this manuscript therefore represents a very timely contribution. The article is written in a clear and compact style and the methods transparently describe all experimental steps, assuring reproducibility. The study moreover compares prime editing to 'base editing' and HDR techniques and shows broad applicability as well as potential weaknesses of the method in 3D organoids. Because prime editing has been described, what I am lacking is a side-by-side comparison with 2D cell lines, e.g. on pegRNA efficacies and editing outcomes. This would be important because it can help to determine to which extent results from heterologous 2D systems can be transferred to the much more demanding 3D organoid system (in particular for Fig 1G and Fig 2 J).

In addition to this point I have a number of specific comments:

The abstract is little informative and should include the following information to increase its scope: gene names of (some of) the modified loci, statement on efficacy compared to HDR, BE and 2D cell lines.

The comparison in Fig. 2F between PE3 and HDR is not entirely fair and could be misleading. A high efficacy of correct editing would also be expected with HDR upon FA selection, but this condition has unfortunately not been shown here. In addition, several HDR and PE3 designs should be studied to exclude that the result is due to one sub-optimal gRNA or HDR template.

Off-target editing should at least be discussed.

The data in Fig S6 is essential and should be added to the main manuscript.

Reviewer #2:

Remarks to the Author:

The recently-developed prime editor (PE) uses an engineered Cas9 nickase–reverse transcriptase fusion protein paired with an engineered prime editing guide RNA (pegRNA) to produce desired genetic variations. In this study, Schene et al report a successful prime editing system in patient-derived liver and intestinal stem cell organoids. They first optimize the prime editing for organoid cells and provide a protocol for prime editing in human adult stem cells. They then demonstrate that prime editing can generate small insertions, small deletions, and various point mutations at several endogenous target sites, and functionally correct several disease phenotypes in patient-derived stem cells. Moreover, they compared the prime editing with HDR and base editing, and find that PE is more efficient and produced lower indels than HDR, but less robust than ABE. In summary, the study presents here the first report of using PE to generate targeted base conversion mutations, small insertions and deletions in primary adult stem cells. These data support the clinical potential of PEs in correcting a broad range of mutations for genetic diseases. I have some comments to improve the manuscript:

1. Did the authors observe the editing byproducts (such as indels or pegRNA scaffold insertions or replacements) of prime editing in their study, like in the human cancer lines and plants? And what is the ratio of the PPE editing byproducts to all editing events?
2. Did the authors analyze unintended effects, such as genome-scale examination of off-target

editing and a deeper understanding of cellular impact (e.g., induced stress responses and adverse growth effects across cell types) in the work?

3. I am confused with the data presented in Supplementary Data (Excel) and have some questions:

1) In the sheet of Fig.2d, there are PE3, PE3+m, PE3b and PE3b+m treatments, but in the main text and figure 2d, there are no such treatments, and instead they were S210d S210d and 2xS210d+PE3 treatments. They were not consistent. The authors have to clarify it.

2) In the sheet of Fig.2j, CuCl should be CuCl₂. In the column of KO2, there are only two sets of data, not three.

3) There is no citation of this excel in the manuscript.

4. There are some minor comments about the figure and citation of figures and references:

1) In the figure 1b, the authors should give the original Sanger sequencing chromatograms for the 5bp deletion in HEK3, because the present chromatogram can mislead that the mutation is homozygous 5bp deletion and the mutation frequency is 100%.

2) In the figure 1g and 2d, what "NC" stands for should be clarified.

3) In the figure 2j, it is better to display the brightfield images after PE3 with pegRNA3 not only with pegRNA1 and pegRNA2.

4) In the figure 2f, it is not clear whether the authors did the experiments of HDR with (FA) OA selection.

5) Line 116, the citation of Figures 1G and 2J should be Figures 1g and 2j.

6) Line 47, the reference 4 should be cited.

7) In the reference 4, Anzalone et al. (2019) demonstrated that prime editing works also in one primary cell type (mouse cortical neurons), not only in the human cancer cells, so the correct information should be added in the text.

Reviewer #1 (Remarks to the Author):

The manuscript by Schene and Joore describes adaptation of the “prime editing” method for human adult stem cell derived organoids. This provides an exciting and versatile new possibility for gene correction and gene targeting in primary human epithelia. The methodology has only reported last year and this manuscript therefore represents a very timely contribution. The article is written in a clear and compact style and the methods transparently describe all experimental steps, assuring reproducibility. The study moreover compares prime editing to ‘base editing’ and HDR techniques and shows broad applicability as well as potential weaknesses of the method in 3D organoids. Because prime editing has been described, what I am lacking is a side-by-side comparison with 2D cell lines, e.g. on pegRNA efficacies and editing outcomes. This would be important because it can help to determine to which extent results from heterologous 2D systems can be transferred to the much more demanding 3D organoid system (in particular for Fig 1G and Fig 2 J).

We thank the reviewer for the kind words and appreciation of the timeliness of this study. We agree that a side-by-side comparison between human stem cell organoids and 2D cell lines is a valuable addition. We therefore included this comparison for two prime edit conditions that were already known to achieve reasonable efficiency in organoid cells: a 5-bp deletion in HEK3 (Fig. 1b) and a 6-bp deletion in CTNBN1 (Fig 1g). Using high-throughput sequencing of alleles from transfected cells, we show similar editing efficiency in liver and intestinal organoid cells compared to HEK293T and Caco-2 cells (Fig. 1h). High-throughput sequencing additionally allowed quantification of rare, unwanted editing byproducts of prime editing, such as indels but also base substitutions. We find that these unwanted byproducts occur at a low rate (<5% of all sequenced alleles) around the pegRNA nicking site and at a very low rate (<1% of all sequenced alleles) around the nicking sgRNA nicking site. Moreover, the rate and pattern/sequence of unwanted byproducts is similar in all tested cell types. We conclude that the efficiency and accuracy of prime editing is similar in 3D grown organoids compared to 2D grown cell lines.

We could not include a 2D-side-by-side comparison of the prime edits performed in figure 2 (including ATP7B in Fig. 2j = Fig. 2k in the new version) because these edits correct pathogenic mutations in patient-derived stem cell organoids. No 2D grown cell lines were available with the identical pathogenic mutations.

In addition to this point I have a number of specific comments:

The abstract is little informative and should include the following information to increase its scope: gene names of (some of) the modified loci, statement on efficacy compared to HDR, BE and 2D cell lines.

This information has been added to the abstract.

The comparison in Fig. 2F between PE3 and HDR is not entirely fair and could be misleading. A high efficacy of correct editing would also be expected with HDR upon FA selection, but this condition has unfortunately not been shown here. In addition, several HDR and PE3 designs should be studied to exclude that the result is due to one sub-optimal gRNA or HDR template.

The HDR template was already an optimized sgRNA and HDR template combination. The other HDR sgRNA tested did not result in any correct HDR events. The misleading PE3+FA condition has been removed from this figure.

Off-target editing should at least be discussed.

We included an in-depth analysis of off-target editing through whole-genome sequencing (WGS) of prime-edited clonal organoid lines (Fig. 3 and Supplementary Fig. 10). To the best of our knowledge, this represents the first genome-wide investigation of off-target effects of prime editors in human cells. We find that prime-editing does not induce mutations in any of the predicted off-target sites (200-500 predicted sites with up to 4 mismatches with either the pegRNA or nicking sgRNA protospacer included).

The data in Fig S6 is essential and should be added to the main manuscript.

The quantification of editing efficiency from Supplementary Fig. S6 has been added to Figure 2g.

Reviewer #2 (Remarks to the Author):

The recently-developed prime editor (PE) uses an engineered Cas9 nickase–reverse transcriptase fusion protein paired with an engineered prime editing guide RNA (pegRNA) to produce desired genetic variations. In this study, Schene et al report a successful prime editing system in patient-derived liver and intestinal stem cell organoids. They first optimize the prime editing for organoid cells and provide a protocol for prime editing in human adult stem cells. They then demonstrate that prime editing can generate small insertions, small deletions, and various point mutations at several endogenous target sites, and functionally correct several disease phenotypes in patient-derived stem cells. Moreover, they compared the prime editing with HDR and base editing, and find that PE is more efficient and produced lower indels than HDR, but less robust than ABE.

In summary, the study presents here the first report of using PE to generate targeted base conversion mutations, small insertions and deletions in primary adult stem cells. These data support the clinical potential of PEs in correcting a broad range of mutations for genetic diseases. I have some comments to improve the manuscript:

1. Did the authors observe the editing byproducts (such as indels or pegRNA scaffold insertions or replacements) of prime editing in their study, like in the human cancer lines and plants? And what is the ratio of the PPE editing byproducts to all editing events?

High-throughput sequencing of targeted loci was performed to address this point. Unwanted byproducts did occur but at a very low rate (<5%). The ratio of PE byproducts to correct edits was about 1:10-20 for the two considered edits (Fig. 1h).

2. Did the authors analyze unintended effects, such as genome-scale examination of off-target editing and a deeper understanding of cellular impact (e.g., induced stress responses and adverse growth effects across cell types) in the work?

All this information has been added: genome scale off-targets in Fig. 3, cellular impact on organoid reconstitution and absence of adverse growth effect in Supplementary Fig. 5a.

3. I am confused with the data presented in Supplementary Data (Excel) and have some questions:

1) In the sheet of Fig.2d, there are PE3, PE3+m, PE3b and PE3b+m treatments, but in the main text and figure 2d, there are no such treatments, and instead they were S210d S210d and 2xS210d+PE3 treatments. They were not consistent. The authors have to clarify it.

It is indeed true that different pegRNA designs and nicking positions were tested. The figure and text have been adapted to match the data sheet.

2) In the sheet of Fig.2j, CuCl should be CuCl₂. In the column of KO2, there are only two sets of data, not three.

This has been changed, the omitted column has been added. We thank the reviewer for this attentive observation.

3) There is no citation of this excel in the manuscript.

The excel file (Sup Table 5) is now cited in the method section.

4. There are some minor comments about the figure and citation of figures and references:

1) In the figure 1b, the authors should give the original Sanger sequencing chromatograms for the 5bp deletion in HEK3, because the present chromatogram can mislead that the mutation is homozygous 5bp deletion and the mutation frequency is 100%.

To assess the efficiency of a given edit in organoids, Sanger sequencing was always performed in individual clones and not in the bulk of transfected organoid cells. In the previous version we chose to show a biallelic edited clone; it is true that this is quite uncommon using prime editing. For clarity, we now chose to show the Sanger sequence of a -more common- monoallelic edited clone, containing both the WT-allele and the 5-bp deletion allele.

2) In the figure 1g and 2d, what “NC” stands for should be clarified.

This has been clarified in Fig. 1e (for figure 1g) and in figure 2c (for figure 2d). This has also been added to the figure legends.

3) In the figure 2j, it is better to display the brightfield images after PE3 with pegRNA3 not only with pegRNA1 and pegRNA2.

This had been added, but the brightfield image-figure is now fig. 2k.

4) In the figure 2f, it is not clear whether the authors did the experiments of HDR with (FA) OA selection.

This has been clarified in the figure legends, the conditions with OA selection (new Fig. 2c) and without OA selection (new Fig. 2f) have been split between two figures for clarity.

5) Line 116, the citation of Figures 1G and 2J should be Figures 1g and 2j.

This has been changed.

6) Line 47, the reference 4 should be cited.

This has been added.

7) In the reference 4, Anzalone et al. (2019) demonstrated that prime editing works also in one primary cell type (mouse cortical neurons), not only in the human cancer cells, so the correct information should be added in the text.

This has been added.

Reviewers' Comments:

Reviewer #1:

Remarks to the Author:

The authors have carefully addressed all my concerns. In particular, the ngs-based detection of off-target effects and the benchmarking with 2D cell lines make this manuscript a very useful resource. The clarity of data presentation and the information content of the abstract have now been improved and I can therefore support the immediate acceptance of this fantastic paper.

Reviewer #2:

Remarks to the Author:

In the revised manuscript, the authors have performed a suite of new experiments addressing my previous concerns including the analysis of genome scale off-targets, prime editing byproducts and prime editing on cellular impact. There are two remaining small points that the authors should be able to address quickly.

1. I do not understand why the authors chose PE1- and PE2-edited clones to perform the genome-wide scale off-target effects. In this study the authors used PE3 or PE3b all the time. Did the authors use PE1 and PE2 to edit the same targets? I did not find any related information, such as PE1- and PE2-edited clones, in the main text, Supplementary files or methods. In particular, the PE1 has wild-type reverse transcriptase whileas PE2, PE3 and PE3b has mutated reverse transcriptase, which may influence the genome-wide off-target effects. The authors should clarify in the text.

2. Proportion of different type of mutations should be marked in all related pie graphs, which may be more clear and intuitive to understand the editing efficiency of the system. For example, in figure1b, what were the proportion of relative biallelic edit, monoallelic edit, and no edit induced by prime editing?

REVIEWERS' COMMENTS

Reviewer #1 (Remarks to the Author):

The authors have carefully addressed all my concerns. In particular, the ngs-based detection of off-target effects and the benchmarking with 2D cell lines make this manuscript a very useful resource. The clarity of data presentation and the information content of the abstract have now been improved and I can therefore support the immediate acceptance of this fantastic paper.

We thank the reviewer for these kind words and we are happy that the reviewer recognizes the added value of the NGS-based off-target screen.

Reviewer #2 (Remarks to the Author):

In the revised manuscript, the authors have performed a suite of new experiments addressing my previous concerns including the analysis of genome scale off-targets, prime editing byproducts and prime editing on cellular impact. There are two remaining small points that the authors should be able to address quickly.

1. I do not understand why the authors chose PE1- and PE2-edited clones to perform the genome-wide scale off-target effects. In this study the authors used PE3 or PE3b all the time. Did the authors use PE1 and PE2 to edit the same targets? I did not find any related information, such as PE1- and PE2-edited clones, in the main text, Supplementary files or methods. In particular, the PE1 has wild-type reverse transcriptase whileas PE2, PE3 and PE3b has mutated reverse transcriptase, which may influence the genome-wide off-target effects. The authors should clarify in the text.

The WGS was not performed on samples with PE1 or PE2, but only with PE3. We understand the naming of the duplicate WGS samples (PE1 and PE2) was confusing and we therefore changed this to PE3-1 and PE3-2.

2. Proportion of different type of mutations should be marked in all related pie graphs, which may be more clear and intuitive to understand the editing efficiency of the system. For example, in figure 1b, what were the proportion of relative biallelic edit, monoallelic edit, and no edit induced by prime editing?

The proportion of different mutations has been added as text to the pie chart for more immediate and intuitive interpretation of efficiency.